# Data-Driven Parameter Estimation of Nonlinear Ship Manoeuvring Model in Shallow Water Using Truncated Least Squares Support Vector Machines

Haitong Xu *📵 and C. Guedes Soares 📵

Centre for Marine Technology and Ocean Engineering (CENTEC), Instituto Superior Técnico,
Universidade de Lisboa, Av. Rovisco Pais, 1049-001 Lisboa, Portugal; c.guedes.soares@centec.tecnico.ulisboa.pt
* Correspondence: haitong.xu@centec.tecnico.ulisboa.pt

**Abstract:** A data-driven method, the truncated LS-SVM, is proposed for estimating the nondimensional hydrodynamic coefficients of a nonlinear manoeuvring model. Experimental data collected in a shallow water towing tank are utilized in this study. To assess the accuracy and robustness of the truncated LS-SVM method, different test data sizes are selected as the training set. The identified nondimensional hydrodynamic coefficients are presented, as well as the corresponding parameter uncertainty and confidence intervals. The validation is carried out using the reference data, and statistical measures, such as the correlation coefficient, centred RMS difference, and standard deviation are employed to quantify the similarity. The results demonstrate that the truncated LS-SVM method effectively models the hydrodynamic force prediction problems with a large training set, reducing parameter uncertainty and yielding more convincing results.

**Keywords:** data-driven; parameter estimation; large-scale training set; truncated LS-SVM; shallow water

## 1. Introduction

The numerical simulation of marine surface ships has played an increasingly important role in modern maritime engineering and design; it can be used for ship manoeuvrability prediction [1,2], safety evaluation [3], and ship operation simulators [4], which benefit from the fast development of computer technology and ship manoeuvring theory [5,6]. This innovative technique involves utilizing computational models and algorithms to replicate the complex behaviour of ships in various environmental conditions, such as waves, wind, and currents. The performance, stability, and safety of marine surface ships can be assessed easily, without the need for expensive physical prototypes and extensive sea trials. Moreover, it facilitates the study of emergency scenarios, aiding in the development of robust safety measures. For example, it can be observed that the size and number of ships are increasing in harbours, and the heavy traffic conditions require the operators to be very careful in steering the marine surface ships [7], which inevitably gives a high requirement for the prediction of the manoeuvring characteristics of ships in shallow water. Several works can be found on such topics such as [8–13] just to name a few.

The nonlinear manoeuvring models are typically used in simulators of marine ships, such as the Abkowitz model [14,15] and its revised version [16–19], the MMG model [20–22], and the vectorial model [23–25]. The manoeuvring model is based on a set of equations that consider various factors affecting a ship's turning performance. These factors include the ship's hull form, propulsion system, rudder characteristics, and environmental conditions such as wind and current. Those manoeuvring models are developed by approximating forces and moments using a set of specific hydrodynamic terms, which may vary between different ships. They are usually determined by the ship hull characters, speed, and environmental conditions and have different values for specific ships. Consequently, when

dealing with a manoeuvring model, the primary objective is to determine the values of the hydrodynamic coefficients associated with these terms.

The most reliable way is to directly measure the values using the ship model in a towing tank [6,26–28], but those tests are usually expensive and time-consuming, and only linear terms can be directly measured. Besides the towing tests in the laboratory, there are also other ways to investigate the manoeuvrability of surface ships, such as sea trials and scaled ship model tests. The free-running ship model test is a promising alternative solution [13,29–33], which is much cheaper than full-scale tests [34–36]. Suzuki et al. [37] conducted a study on the manoeuvrability of a tank ship, utilizing ship model tests for validation. Free-running model tests were employed to assess the impact of shallow water on the manoeuvring behaviour of the very large crude carrier KVLCC2 [29]. System identification methods were used to estimate the ship manoeuvring mathematical models using simulation data and free-running tests.

System identification is a crucial process in the field of engineering, and it involves the estimation and characterization of mathematical models that represent the dynamic behaviour of complex systems. It can be used to extract information about a system's behaviour by observing its inputs and outputs. Therefore, system identification also plays an important role in building the mathematical model for marine vessels [38–42]. This involves conducting experiments and analysing ship manoeuvring test data to determine the relationships between inputs and outputs, typically in the form of mathematical models or transfer functions. Åström and Källström [41] applied the system identification techniques to obtain the parameters of ship steering dynamics. The Least squares is an important method and was widely used for various applications [43–45]. Wang et al. [46] proposed a hybrid recursive least squares method for online identification.

Qian et al. [47] proposed an optimized deep long short-term memory network framework (LSTM) to predict the ship trajectory of inland water, and the experimental results showed that the GA-LSTM model can effectively improve the accuracy and speed of trajectory prediction. An offline genetic algorithm was used to estimate the ship's manoeuvrability using CFD simulations of free-running model tests [48]. Xu and Guedes Soares [49] discussed the parameter error and convergence problem of the hydrodynamic coefficients estimation of a nonlinear manoeuvring model. Wang et al. [50] proposed a generalized ellipsoidal function-based fuzzy neural network (GEBF-FNN) to describe the reference model for a large tanker. The obtained models were used to simulate the typical zig-zag manoeuvres with moderate and extreme steering. Dong et al. [51] proposed a math-data integrated prediction (MDIP) model for ship manoeuvring motion, where the variable-order hydrodynamic derivatives were used. The results show that the proposed model can offer a stronger generalization, and possibly be used for the ship manoeuvring motion prediction.

Recently, a kernel-based machine learning method, support vector machine (SVM), has been used for the manoeuvring modelling of ships, considering its various advantages [15,38–40,52–54] such as relatively memory efficient, global and unique solution, and sparseness [55–59]. However, there are also some disadvantages. One of them is that SVM is not suitable for large data sets. As indicated in [60], the size of the training set for the LS-SVM should be limited to fewer than 2000 data points. A limited training set makes it not suitable for the data-driven modelling problem of complex systems, for example, the manoeuvring modelling of large container ships in shallow water, since the kinematic theory behind the shallow water effect is still blurred.

Recently, a novel version of the Support Vector Machine (SVM), known as the truncated LS-SVM, was introduced for the analysis of ship manoeuvrability based on Planar Motion Mechanism (PMM) tests, using a large-scale training dataset. The truncated LS-SVM was successfully applied to estimate hydrodynamic coefficients in various conditions, including deep water [24,61], shallow water [62,63], and free-running ship mode tests [13,64]. The size of the training dataset plays a critical role in the performance of parameter estimation methods, and the same holds for the truncated LS-SVM. Consequently, it is imperative

to investigate the impact of training dataset sizes on the determination of hydrodynamic coefficients and the associated parameter uncertainty.

This paper aims to analyse data-driven parameter estimation of a nonlinear manoeuvring model using PMM test data. Additionally, it will validate the performance of the truncated LS-SVM across a range of training set sizes from 1000 to 10000. The paper will discuss parameter uncertainty arising from noise and provide confidence intervals for the identified parameters. The validation will be carried out using the statistical merits of the prediction and reference data, such as the correlation coefficient (R), centred RMS difference, and standard deviation.

## 2. Ship Manoeuvring Model

Surface ships in wave conditions are commonly treated as rigid bodies, with their motion described by 6 Degrees of Freedom (DOF), as shown in Figure 1. These DOFs encompass surge, sway, and yaw (manoeuvring motion), as well as roll, pitch, and heave motion (seakeeping motion). In the context of manoeuvring studies, the focus is typically on coupled motions within the horizontal plane, with an assumption of constant values for frequency-dependent added mass and potential damping [23].

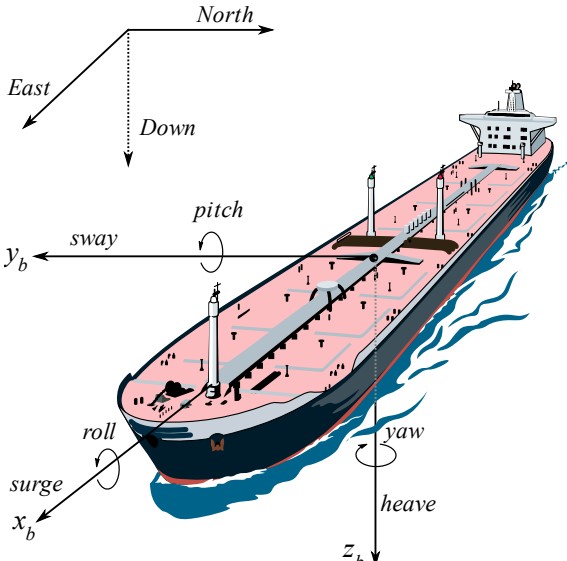

**Figure 1.** Coordinate systems to study the motions of ships in waves.

To describe the manoeuvring motion of the ship, an empirical manoeuvring model is presented in this section. The equations governing the ship's behaviour under the influence of hydrodynamic forces and moments are defined as

$$
\begin{aligned}
(m + X_{\dot{u}})\dot{u} - mvr - mx_G r^2 &= X_q + X_p \\
(m + Y_{\dot{v}})\dot{v} + (mx_G + Y_{\dot{r}})\dot{r} + mur &= Y_q \\
(mx_G + N_{\dot{v}})\dot{v} + (I_{zz} + N_{\dot{r}})\dot{r} + mx_G ur &= N_q
\end{aligned}
\tag{1}
$$

where $m$ and $Izz$ are the mass and inertial moment of the ship, respectively. $X_{\dot{u}}, Y_{\dot{v}}, Y_{\dot{r}}, N_{\dot{v}}, N_{\dot{r}}$ are the added mass coefficients, $x_G$ is the longitudinal coordinate of the centre of mass, and $X_p$ is the surge force induced by a propeller. The quasi-steady hydrodynamic forces and moments on the ship hull and rudder are $X_q, Y_q, N_q$.

In this paper, only the hull forces and moment are considered because the PMM test data were carried out using the bare model ship hull. The dimensionless forces and moment are defined as multi-variate regression polynomials depending on the nondimensional velocities [65], $u' = u/U, v' = v/U, r' = rL/U$.

$$
X_q' = X_0' + X_{uu}' u'u' + X_{vr}' v'r'
\tag{2}
$$

$$Y'_q = Y'_0 + Y'_v v' + Y'_r r' + Y'_{vv} v' |v'| + Y'_{vr} v' |r'| + Y'_{rr} r' |r'| \tag{3}$$

$$N'_q = N'_0 + N'_v v' + N'_r r' + N'_{rv} r' |v'| + N'_{vr} v' |r'| + N'_{r|r|} r' |r'| \tag{4}$$

The above quasi-polynomial regression models are the revised version of the nonlinear manoeuvring models that were proposed by Inoue et al. [66,67]. This model gives a satisfactory agreement with the full-scale trial results and can be used for the prediction of ship manoeuvrability in the initial ship design stage. To describe the shallow water effect on the hydrodynamic forces, it is assumed that the values of the hydrodynamic coefficients in Equations (2)–(4) are related to the shallow water effect. The hydrodynamic forces and moments are nondimensionalized by using the prime system recommended by SNAME [68].

$$X'_q = X_q \Big/ 0.5\rho U^2 LT, Y'_q = Y_q \Big/ 0.5\rho U^2 LT, N'_q = N_q \Big/ 0.5\rho U^2 L^2 T \tag{5}$$

where $\rho$ is the water density, $L$ is the ship length, $U$ is the ship speed over ground, and $T$ is the draught at the midship. The hydrodynamic coefficients in Equations (2)–(4) are dimensionalized using the factors given in Table 1.

**Table 1.** Dimensional factors for the hydrodynamic parameters.

| Coef. | Dimensional Factor | Coef. | Dimensional Factor | Coef. | Dimensional Factor |
|---|---|---|---|---|---|
| $X'_u$ | $0.5\rho L^2 T$ | $Y'_v$ | $0.5\rho LTU$ | $N'_0$ | $0.5\rho L^2 TU^2$ |
| $X'_0$ | $0.5\rho LTU^2$ | $Y'_r$ | $0.5\rho L^2 TU$ | $N'_v$ | $0.5\rho L^2 TU$ |
| $X'_{uu}$ | $0.5\rho LT$ | $Y'_{v|v|}$ | $0.5\rho LT$ | $N'_r$ | $0.5\rho L^3 TU$ |
| $X'_{vr}$ | $0.5\rho L^2 T$ | $Y'_{v|r|}$ | $0.5\rho L^2 T$ | $N'_{r|v|}$ | $0.5\rho L^3 T$ |
| $Y'_{\dot{v}}$ | $0.5\rho L^2 T$ | $Y'_{r|r|}$ | $0.5\rho L^3 T$ | $N'_{v|r|}$ | $0.5\rho L^3 T$ |
| $Y'_{\dot{r}}$ | $0.5\rho L^3 T$ | $N'_{\dot{v}}$ | $0.5\rho L^3 T$ | $N'_{r|r|}$ | $0.5\rho L^4 T$ |
| $Y'_0$ | $0.5\rho LTU^2$ | $N'_{\dot{r}}$ | $0.5\rho L^4 T$ | | |

## 3. Duisburg Model Tested in Shallow Water

The hydrodynamic parameter estimation training dataset comprises planar motion mechanism (PMM) test data employing the Duisburg Test Case (DTC) ship model. The DTC ship model is a well-known and widely used benchmark in the field of ship hydrodynamics and manoeuvring. Many experimental tests were carried out using the DTC ship model and the results serve as a standardized test case for assessing and validating numerical simulation techniques, particularly those related to ship manoeuvring performance. It is used as a fundamental reference point for assessing and advancing the capabilities of numerical simulations in ship manoeuvring. Its standardized geometry and parameters make it an invaluable tool for improving the accuracy of ship design and performance prediction methods, ultimately benefiting the maritime industry as a whole.

The DTC model tests were carried out under the support of the SHOPERA project [69–71]. The main reason for using the PMM test is the quality of the data. The PMM test is a critical experimental method used to assess and characterize the hydrodynamic behaviour of ship models. This test involves a specialized apparatus known as a PMM that allows for controlled and precise movement of the ship model in a testing tank, simulating different types of ship motions. During the PMM ship test, the model is placed in a large water tank, and the PMM system precisely controls its movements. The model is subjected to various input motions, replicating the effects of waves, wind, and other environmental forces. The data on how the ship model responds to these simulated conditions, including its resistance, stability, and motion characteristics can be collected using data acquisition instruments.

In this paper, the PMM tests were conducted in a towing tank under shallow water conditions at Flanders Hydraulics Research (FHR), where measurements of hydrodynamic forces and moments acting on the bare hull were acquired. The quality of the test data used in this paper is very high and reliable, and to some extent, this can reduce the uncertainty due to noise.

The towing tank at FHR measures 87.5 m in length, 7 m in width, and has a maximum water depth of 0.5 m, rendering it suitable for conducting model tests in both shallow and very shallow water conditions. More detailed information can be found in [72,73]. Figure 2 displays the DTC ship model positioned on the carriage within the towing tank during testing, while Table 2 provides the key dimensions of the ship model.

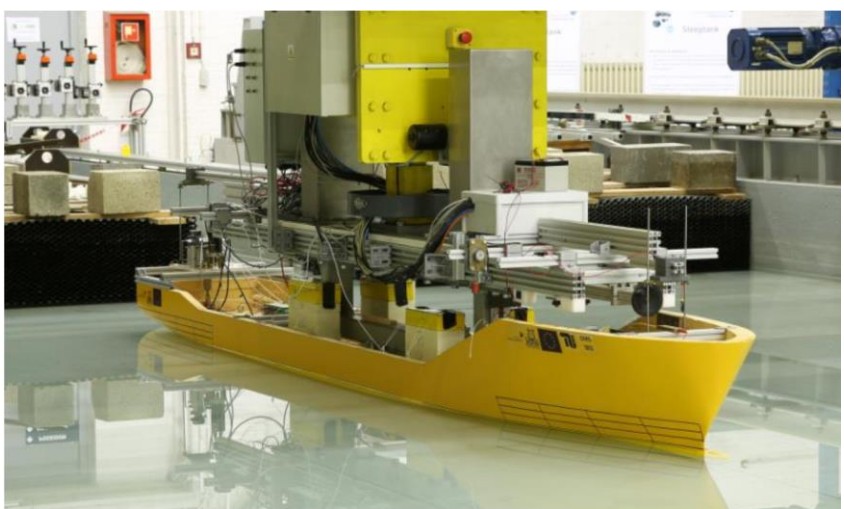

**Figure 2.** Planar motion mechanism tests of DTC ship model in shallow water. Reprinted from Ref. [63] with permission from Elsevier, 2023.

**Table 2.** The dimensions of DTC model (1:89.11).

| Description | |
| --- | --- |
| Length between pp ($L_{pp}$) | 3.984 m |
| Draught ($T$) | 0.163 m |
| Beam ($B$) | 0.572 m |
| Block coefficient ($C_b$) | 0.661 |
| Mass | 242.8 kg |
| centre of gravity in $x$-direction ($x_G$) | −0.052 m |
| Moment of inertia along $z$-axis ($I_{zz}$) | 219 kg m$^2$ |

The 60 PMM test cases were executed in the towing tank with a water depth, of 0.3254 m (the water depth to draught ratio h/T is 2). The raw results of all model tests were 40 Hz time series, and the four force gauges were installed on the towing platform. The surge, sway forces, and yaw moments were calculated based on the measured signals of the four separate force gauges. The tests included the pure drift, pure sway, and coupled sway–yaw test. To fully activate the response of the ship and obtain rich information, the velocities of towing speed and the amplitude and frequency of the oscillatory motion in tests were changed during the tests; for example, 3 different speeds, 7 drift angles, 3 amplitudes, and 2 frequencies were considered. The PMM tests are described in Figure 3. As can be observed, the pure drift was carried out using the ship model with a fixed drift angle. The pure sway test is the ship model oscillated around the y-axis with zero drift angle. During the pure yaw test, the ship model moved forward with a sinusoidal oscillation in the y-axis with zero sway speed, as presented in Figure 3c. The coupled sway–yaw test was the pure yaw test with no zero-drift angle.

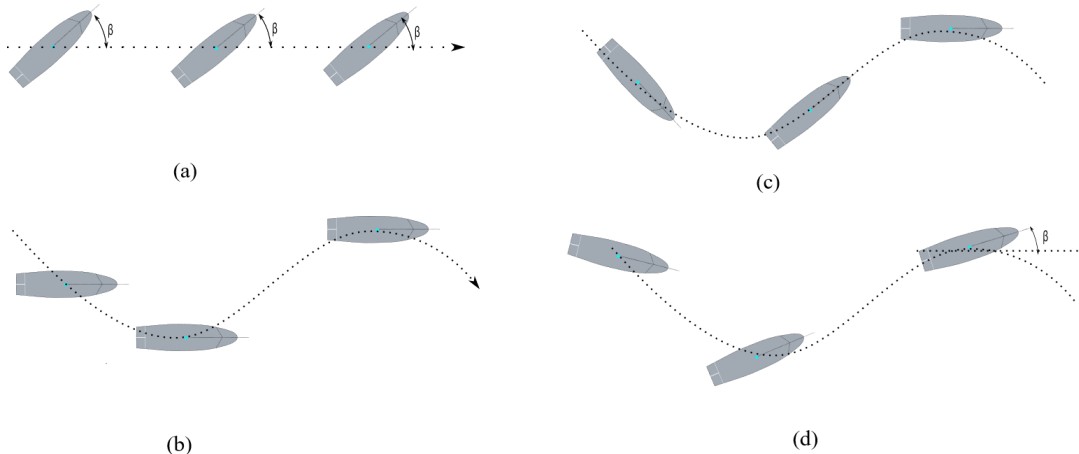

**Figure 3.** Ship model towing tests: (**a**) pure drift; (**b**) pure sway; (**c**) pure yaw; and (**d**) pure yaw +drift.

## 4. Truncated LS-SVM

Least Squares Support Vector Machines (LS-SVM) are a supervised learning algorithm that extends the original concept of SVM from classification to regression tasks. SVMs have a good performance for classification problems by finding a hyperplane that best separates two classes while maximizing the margin between them. LS-SVM adapts this idea to regression problems where the goal is to predict continuous numerical values.

The main objective of LS-SVM is to find a hyperplane that best fits the data by minimizing the regression error. It focuses on minimizing the error between the actual target values and the predicted values along with a regularization term. The regularization term helps prevent overfitting. The kernel function is also used in LS-SVMs, where it can map data into a higher-dimensional feature space, making them capable of handling non-linear relationships in the data.

This section introduces a novel iteration of the support vector machine, referred to as the truncated LS-SVM, and delves into the parameter uncertainty resulting from data noise. The classical LS-SVM was proposed by [60], and it is obtained by reformulating the minimization problem using the regression errors, as presented in [60].

One of the significant advantages of LS-SVM is that it achieves results by solving a set of linear equations, as opposed to the quadratic programming (QP) problems typically associated with classical SVMs. It can simplify the required computation, but unfortunately, the sparseness of standard SVM is lost. Therefore, the classical LS-SVM is not recommended for large-scale data applications, or more precisely, for large-scale training problems. As recommended by [60], the size of the training set is usually restricted to about N = 2000. In the following part, the truncated LS-SVM will be proposed for the manoeuvring modelling with the large-scale training set. The uncertainty of the identified parameters is also analysed. The classical LS-SVM is given as follows:

$$\underbrace{\begin{bmatrix} 0 & \vec{1} \\ \vec{1} & K(\cdot) + C^{-1}\boldsymbol{I} \end{bmatrix}}_{A} \underbrace{\begin{bmatrix} b \\ \vec{\alpha} \end{bmatrix}}_{\theta} = \underbrace{\begin{bmatrix} 0 \\ \vec{Y} \end{bmatrix}}_{Y} \tag{6}$$

where $\boldsymbol{I}$ is an identity matrix of size $N$, $\vec{\alpha} = [\alpha_1, \cdots \alpha_N]^T$ are the Lagrange multipliers, $\vec{Y} = [y_1, \cdots, y_N]^T$ is the output vector, and $K(x_k \cdot x_i) = \varphi(x_k)^T \varphi(x_i)$, $i = 1, \cdots, N$ is the kernel function, which is positive definite and satisfies the Mercer condition [74]. To estimate the values of the hydrodynamic coefficients, the linear kernel function is chosen. As can be observed in Equation (6), the dimension of matrix $A$ increases exponentially with the size of the training set, which will result in unstable solutions. The obtained parameters are usually sensitive to noise and drift from the true values.

In the following part, the singular values decomposition is introduced for the kernel matrix analysis, and it is given as

$$A = \sum_{i=1}^{n} u_i \sigma_i v_i^T = U\Sigma V^T \tag{7}$$

Then, substituting into Equation (6) gives

$$\theta = \left( U\Sigma V^T \right)^{-1} Y = V\Sigma^{-1} U^T Y = \sum_{i=1}^{n} \frac{v_i u_i^T}{\sigma_i} Y \tag{8}$$

where the matrix, $U$, is orthogonal with the eigenvectors chosen from $AA^T$, and the matrix, $V$, is orthogonal and its eigenvectors are chosen from $A^T A$. $\Sigma$ is a diagonal matrix.

Assume that the output data, $y$, contains the noise, $\delta y$, then the noise will propagate to a perturbation in the solution in Equation (8):

$$\hat{\theta} \doteq \theta_{true} + \delta\theta = \left( U\Sigma V^T \right)^{-1} (y_{true} + \delta y) \tag{9}$$

Then, the perturbation in the solution due to the noise can be obtained as follows:

$$\delta\theta = V\Sigma^{-1} U^T \delta y = \sum_{i=1}^{n} \frac{v_i u_i^T}{\sigma_i} \delta y \tag{10}$$

As presented in Equation (10), and with the discrete Picard condition [75], the portion of the singular values can be kept when the ill-conditioned matrix is obtained from the measured data. The data noise can be magnified and potentially dominate the solutions when the singular values are small. Therefore, to diminish the error propagation due to the noise, it is preferred to neglect the smaller singular values in the matrix $\Sigma$. The matrix can be presented as

$$A_r = U_r \Sigma_r V_r^T \tag{11}$$

The truncated value, $r$, plays a trade-off role between the size of the regularized solutions and their fit to the given data, and the L-curve [76] can be used to obtain the optimal value.

To quantify how random measurement errors in the data, $y$, propagate to the identified parameters, the error propagation matrix can be calculated using

$$V_{\hat{\theta}} = \left[ \frac{\partial \hat{\theta}}{\partial y} \right] V_y \left[ \frac{\partial \hat{\theta}}{\partial y} \right]^T \tag{12}$$

The standard error of the parameters, $\sigma_{\hat{\theta}}$, is the square root of the diagonal of the error propagation matrix. Then, the confidence intervals for the identified parameters are given as follows:

$$\hat{\theta} - t_{(1-a/2)} \ \sigma_{\hat{\theta}} \leq \theta \leq \hat{\theta} + t_{(1-a/2)} \ \sigma_{\hat{\theta}} \tag{13}$$

where $1 - a$ is the desired confidence level, and $t$ is the Student's t statistic. Typically, for the large-scale training set where the number of the measured data is much larger than the number of the estimated parameters, $t$ is 1.96 for 95% confidence intervals and 1.28 for 80% confidence intervals.

With the identified models, it is necessary to validate the models by comparing the prediction with the new test data, which was not used in the training process. Several statistical merits are used to qualify their similarity. Given reference data, $y = [y_i y_2, \cdots, y_N]^T$

and the prediction data, $\hat{y} = [\hat{y}_i, \hat{y}_2, \cdots, \hat{y}_N]^T$, the correlation coefficients can be calculated as follows:

$$R = \frac{1}{N} \sum_{n=1}^{N} (y_i - \overline{y})(\hat{y}_i - \overline{\hat{y}})/\sigma_y \sigma_{\hat{y}} \tag{14}$$

where, $\overline{y}$ and $\overline{\hat{y}}$ are the mean values of the reference data and prediction data, respectively. $\sigma_y$ and $\sigma_{\hat{y}}$ are the standard deviations, and are calculated as follows:

$$\sigma_y = \frac{1}{N} \sum_{n=1}^{N} (y_i - \overline{y}), \ \sigma_{\hat{y}} = \frac{1}{N} \sum_{n=1}^{N} (\hat{y}_i - \overline{\hat{y}}) \tag{15}$$

The centred root-mean-square (RMS) difference is given below.

$$E'^2 = \frac{1}{N} \sum_{n=1}^{N} [(y_i - \overline{y})(\hat{y}_i - \overline{\hat{y}})]^2 \tag{16}$$

## 5. Data-Driven Manoeuvring Modelling of DTC Model

In this section, the proposed system identification method, the truncated LS-SVM, is employed to estimate the nondimensional hydrodynamic coefficients using the PMM tests in shallow water. To test the performance of the proposed truncated LS-SVM for large-scale data-driven modelling, the different training set sizes are considered for the surge, sway, and yaw models. The training set is defined as Surge_{*ID*}, Sway_{*ID*}, and Yaw_{*ID*}, where ID represents the number of data points, as presented in Table 3. As suggested by [60], the data size should be restricted below 2000 because kernel matrix size grows with the number of data points. In this study, the size of the training data is set in different ranges from 1200 to 10,000, and the training data contains the pure drift, pure sway, pure yaw, and coupled sway and yaw tests, as described in Section 3.

**Table 3.** Training set size for the data-driven parameter estimation.

| ID | 1 | 2 | 3 | 4 | 5 | 6 | 7 |
|---|---|---|---|---|---|---|---|
| Training set size | 1200 | 1300 | 1500 | 1700 | 2000 | 2300 | 2900 |
| ID | 8 | 9 | 10 | 11 | 12 | 13 | 14 |
| Training set size | 4000 | 5000 | 6000 | 7000 | 8000 | 9000 | 10,000 |

The truncated LS-SVM is used to identify the nondimensional hydrodynamic coefficients of the ship hull, and the results are presented in Figure 4. As can be observed, the parameters converge to a constant value as the training set size grows. When the training set is around 4000, the obtained results change slightly, even while the training set size still grows. The 80% and 95% confidence intervals of the identified parameters are also presented in the figure. The confidence intervals of the identified parameters represent the theoretical long-run frequency of confidence intervals that contain the true values.

From Figure 4, the confidence intervals decrease with the training set size, which indicates that the margin of error decreases. In plain words, the large scale of the training set can provide more confidence and robust results. Since the noise in data is randomly recorded during data collection due to the device and environmental disturbance, the proposed method can diminish the noise effect, to a certain extent, by using the large-scale training set.

The truncated LS-SVM is also employed for parameter estimation of the nondimensional hydrodynamic coefficients of the nonlinear sway model, as given in Equation (3). There are eight parameters to be estimated, and the training set size was set as indicated in Table 3. The obtained values of the nondimensional coefficients are presented in Figure 5, as well as the confidence intervals of the corresponding ones.

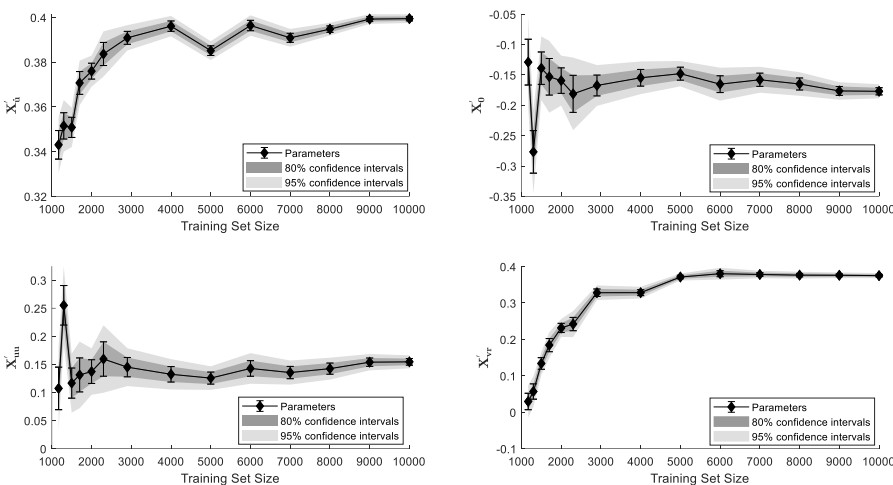

**Figure 4.** The identified hydrodynamic coefficients of the surge model with 80% and 95% confidence intervals.

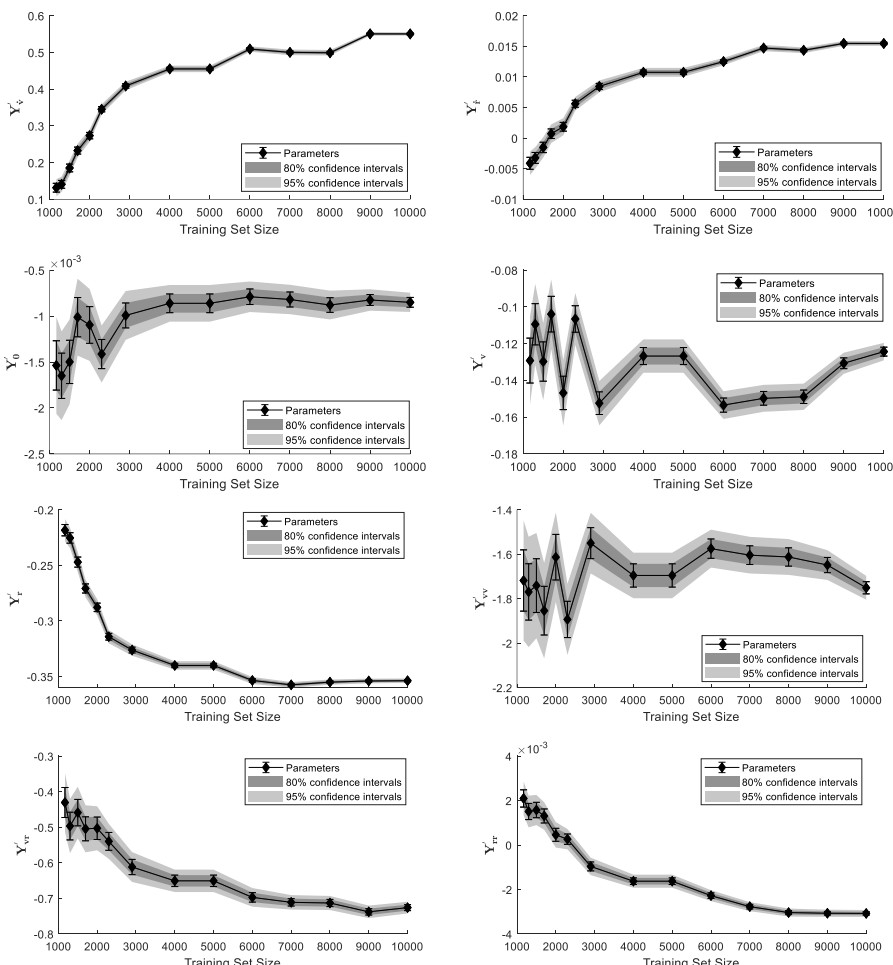

**Figure 5.** The identified hydrodynamic coefficients of the sway model with 80% and 95% confidence intervals.

As can be observed in Figure 5, the parameters can converge to a constant value as the training set size grows, except the parameters $Y_v$, $Y_{vv}$. The $Y_{vv}$ enters a stable period when the training set is around 6000 but decreases slightly when the training set continues to grow, and vice versa for $Y_v$ [52,77]. This can be attributed to the dynamic cancellation,

which results from the multicollinearity of the two terms. The 80% and 95% confidence intervals of the identified parameters are also presented in the figure. From Figure 5, the confidence intervals decrease with the training set size, which also indicates that the large-scale training set can diminish the parameter uncertainty.

The nondimensional hydrodynamic coefficients of the yaw model are identified and presented in Figure 6. The results can converge to a constant value as the training set size grows, as shown in Figure 6. The confidence intervals are also given in the figure, and they decrease with the training set size, which indicates that the uncertainty has been diminished.

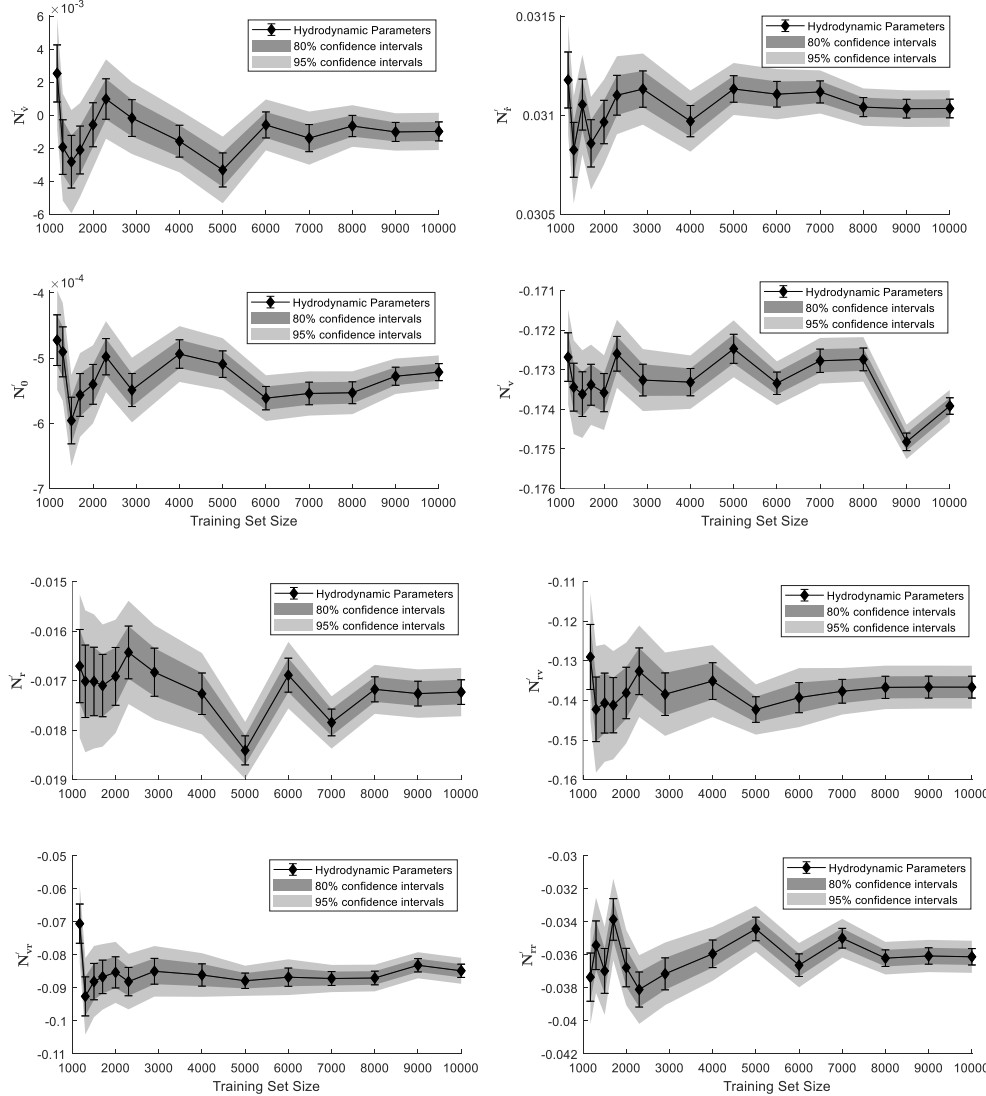

**Figure 6.** The identified hydrodynamic coefficients of the yaw model with 80% and 95% confidence intervals.

To validate the results, which are obtained using the different training set sizes, the models are employed to reproduce the hydrodynamic forces and moments that were measured during the tests. The harmonic yaw and sway test are selected as the test data, and the Taylor diagram [78] is used to show how closely the prediction matches the observations (experimental data). The similarity is quantified in terms of their correlation, root-mean-square differences, and amplitude of variations.

As presented in Figure 7, the statistical merits of the test data are indicated using the red line, and the values of the correlation coefficient (*R*), centred RMS difference, and standard deviation are presented in Table 4. From Figure 7b, the correlation coefficients

of yaw models are very high, and very close to the reference data for all the cases, even when the training set is small. It indicates that the designed PMM tests fully activate the response of yaw motion and are suitable for parameter estimation of yaw motion.

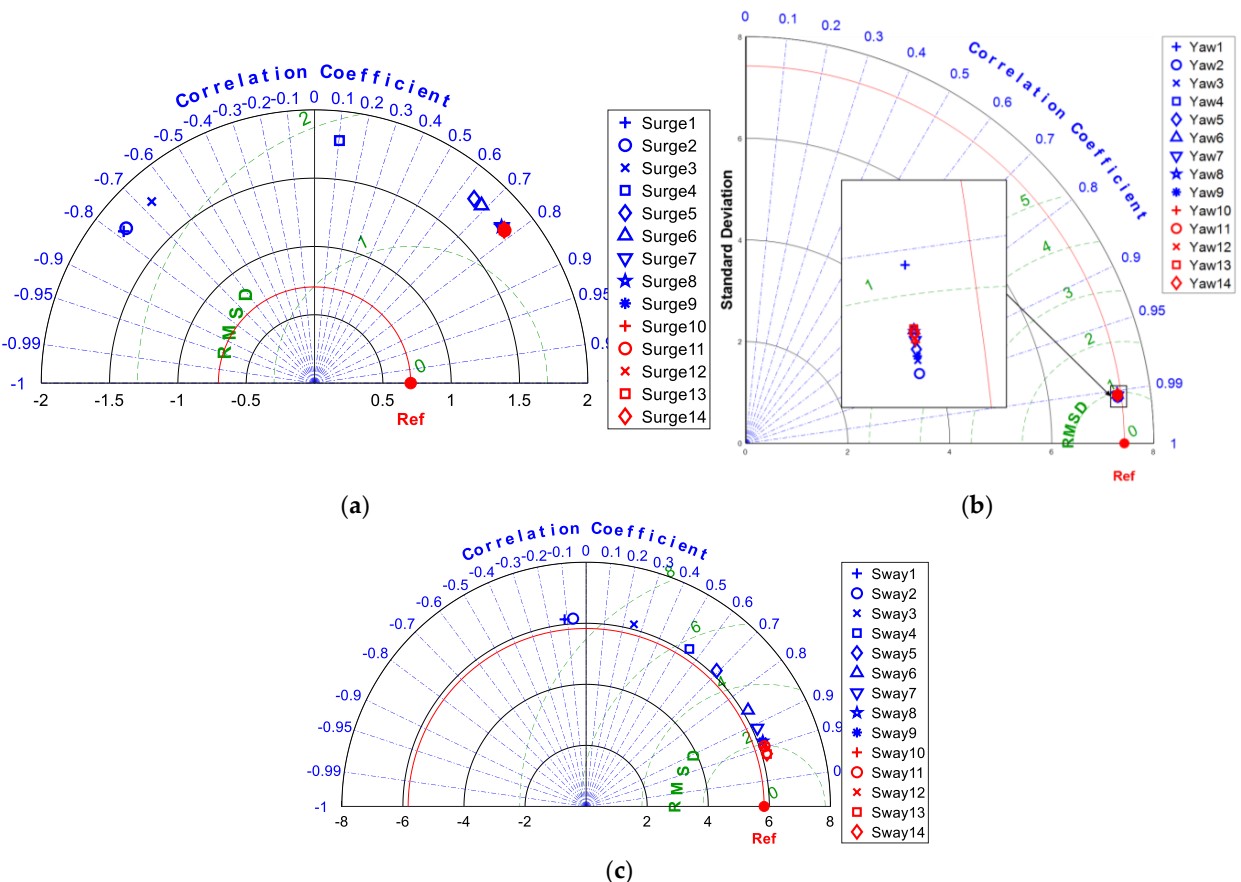

**Figure 7.** Taylor diagram showing the validation performance of the obtained models: (**a**) surge model; (**b**): yaw model; and (**c**): Sway model.

**Table 4.** The statistical merits of the predictions of the obtained surge, sway, and yaw models.

| ID | SURGE | | | SWAY | | | YAW | | |
|---|---|---|---|---|---|---|---|---|---|
| | *R* | *E′* | *STD* | *R* | *E′* | *STD* | *R* | *E′* | *STD* |
| **REF.** | 1.000 | 0.000 | 0.704 | 1.000 | 0.000 | 5.829 | 1.000 | 0.000 | 7.422 |
| **1** | −0.783 | 2.232 | 1.785 | −0.114 | 6.734 | 6.168 | 0.990 | 1.027 | 7.352 |
| **2** | −0.772 | 2.133 | 1.785 | −0.069 | 6.598 | 6.168 | 0.992 | 0.914 | 7.352 |
| **3** | −0.668 | 1.919 | 1.785 | 0.254 | 5.984 | 6.168 | 0.992 | 0.943 | 7.352 |
| **4** | 0.102 | 1.776 | 1.785 | 0.548 | 5.316 | 6.168 | 0.992 | 0.956 | 7.352 |
| **5** | 0.654 | 1.648 | 1.785 | 0.693 | 4.778 | 6.168 | 0.992 | 0.944 | 7.352 |
| **6** | 0.685 | 1.619 | 1.785 | 0.860 | 3.802 | 6.168 | 0.992 | 0.962 | 7.352 |
| **7** | 0.768 | 1.412 | 1.785 | 0.909 | 3.032 | 6.168 | 0.992 | 0.961 | 7.352 |
| **8** | 0.767 | 1.413 | 1.785 | 0.938 | 2.472 | 6.168 | 0.992 | 0.958 | 7.352 |
| **9** | 0.779 | 1.322 | 1.785 | 0.938 | 2.472 | 6.168 | 0.992 | 0.950 | 7.352 |
| **10** | 0.779 | 1.304 | 1.785 | 0.949 | 2.017 | 6.168 | 0.992 | 0.956 | 7.352 |
| **11** | 0.779 | 1.309 | 1.785 | 0.947 | 2.083 | 6.168 | 0.992 | 0.962 | 7.352 |
| **12** | 0.778 | 1.312 | 1.785 | 0.947 | 2.091 | 6.168 | 0.992 | 0.954 | 7.352 |
| **13** | 0.778 | 1.312 | 1.785 | 0.961 | 1.719 | 6.168 | 0.992 | 0.964 | 7.352 |
| **14** | 0.778 | 1.314 | 1.785 | 0.960 | 1.730 | 6.168 | 0.992 | 0.960 | 7.352 |

For the surge and sway motion (Figure 7a,c), the correlation coefficients are negative when the training set size is small, which indicates that the obtained models are negatively related to the test data. In this case, the obtained model cannot be used to predict the surge and sway forces on the ship hull. With the training set size growing, the correlation coefficients can achieve a good level. The centred RMS difference ($E'$) is also presented in Figure 7 by using green contours. For the surge and sway models, the centred RMS differences decrease with the training set size, and it can be found graphically that the markers are close to the Ref. in Figure 7a,c.

It is necessary to point out that the training set size does not change the standard deviation of the prediction of the obtained models, it largely depends on the structure of the nonlinear manoeuvring models. The standard deviation of the sway model agrees very well with the reference data but is greater in the surge case. It also can be confirmed by Figure 8, which presents the predicted surge, sway forces, and yaw moments during the PMM test. The manoeuvring models obtained using the training set size (ID 14 with 10,000 samples) are chosen for validation. From Figure 8, it can be observed that there are more oscillations in the predictions of the surge and sway model.

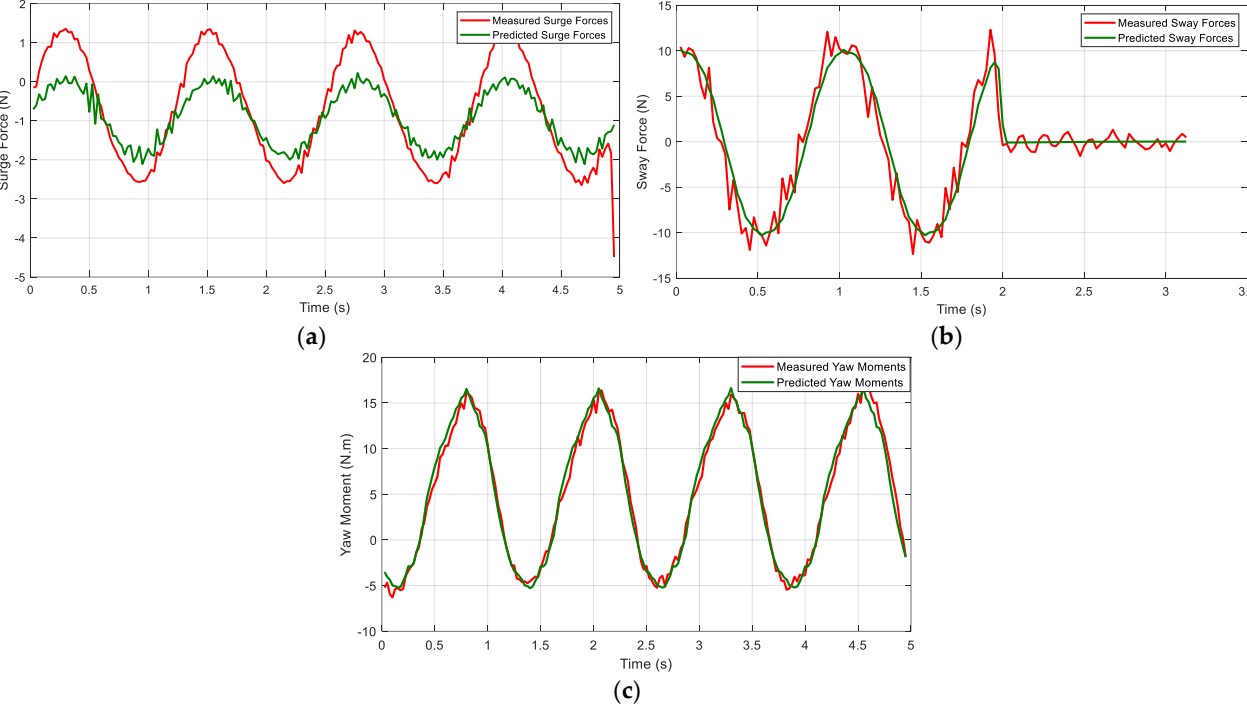

**Figure 8.** Reproduction of the hydrodynamic hull forces and moment: (**a**) Pure yaw with constant drift angle; (**b**): pure sway; and (**c**): harmonic yaw test in shallow water using the obtained models.

## 6. Conclusions

This paper investigates the data-driven parameter estimation of a nonlinear ship manoeuvring model using the truncated LS-SVM, where the PMM tests are used as the training and validation set. The truncated LS-SVM is employed to estimate the nondimensional hydrodynamic coefficients with the 14 different training set sizes, and the parameter uncertainty due to the noise is also presented, as well as the confidence intervals of the identified parameters.

The results indicate that the truncated LS-SVM is capable of the modelling problem using a large-scale training set. The obtained parameters can converge to the constant values and their uncertainty can be diminished as the training set size grows, as well as the margin of confidence intervals. Therefore, the truncated LS-SVM can diminish the parameter uncertainty and provide a robust result. The validation is also carried out using statistical measures: the correlation coefficient (R), centred RMS difference, and the standard

deviation presented graphically using the Taylor diagram. It can be concluded that the PMM test can fully activate the response of yaw motions and provide rich information for the parameter estimation of the yaw model. The large-scale training set can increase the credibility of results by diminishing uncertainties. With the increase of the training set size for surge and sway models, the obtained models agree well with the reference data. This paper focuses on the prediction of the hydrodynamic forces and moments of the bare ship hull in shallow water under the assumption that the values of the hydrodynamic coefficient are directly affected by the shallow water depth. The hydrodynamic terms related to the rudder, propeller, and their interaction are neglected due to the lack of test data, which is the limitation of this paper. In a future study, it is suggested to carry out the PMM test in shallow water using the hull with rudder and propeller, and the hydrodynamic terms explicitly related to the shallow water features should also deserve more attention. The determination of the optimal values of the parameters for the truncated LS-SVM is also an interesting topic for further investigation.

**Author Contributions:** Conceptualization and methodology, H.X. and C.G.S.; software, validation, formal analysis, investigation, data curation, visualization, and writing—original draft preparation, H.X.; writing—review and editing, H.X. and C.G.S., supervision, project administration and funding acquisition, C.G.S. All authors have read and agreed to the published version of the manuscript.

**Funding:** This work was performed within the Strategic Research Plan of the Centre for Marine Technology and Ocean Engineering, financed by the Portuguese Foundation for Science and Technology (Fundação para a Ciência e Tecnologia—FCT) under contract UIDB/UIDP/00134/2020. The PMM data were collected in the experiments performed during the Project "SHOPERA-Energy Efficient Safe SHip OPERAtion", which was funded by the EU under contract 605221.

**Institutional Review Board Statement:** This study does not involve humans or animals.

**Informed Consent Statement:** Not applicable.

**Data Availability Statement:** Not applicable.

**Conflicts of Interest:** The authors declare no conflict of interest.

## Nomenclature

| | |
|---|---|
| LS-SVM | Least-squares support vector machine |
| RMS | Root mean square |
| MMG | Manoeuvring Modelling Group |
| LSTM | Long short-term memory network |
| GA | Genetic Algorithm |
| CFD | Computational fluid dynamics |
| SVM | Support vector machine |
| PMM | Planar Motion Mechanism |
| DOF | Degrees of Freedom |
| DTC | Duisburg Test Case |
| FHR | Flanders Hydraulics Research |
| QP | Quadratic programming |
| R | Correlation coefficient |

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
