# Peer review of "Data-Driven Parameter Estimation of Nonlinear Ship Manoeuvring Model in Shallow Water Using Truncated Least Squares Support Vector Machines"

_jmse, doi:10.3390/jmse11101865_

Round 1

Reviewer 1 Report

Comments and Suggestions for Authors

In this paper, the main research is divided into the following aspects: verifying the accuracy and robustness of the truncated LS-SVM using test data of different sizes; demonstrating the stability of the parameters and the decrease in the confidence interval range due to the increase in the number of data; quantifying the performance of the parameters by three metrics and verifying the correctness of the parameters by using the surge, sway force, and yaw moments in the PMM test.

1.      In Section 5, Fig 4, all vertical coordinates are replaced with nondimensional parameters.

2.      It would be highly more convenient for the reader if they add a table with all abbreviations and their meaning in the first before the introduction section. It was very difficult to keep track as it is now.

3.      Further enrichment of the article, please refer to the following papers: " Large tanker motion model identification using generalized ellipsoidal basis function-based fuzzy neural networks. IEEE Transactions on Cybernetics, 2015, 45(12): 2732-2743", " Hybrid recursive least squares algorithm for online sequential identification using data chunks. Neurocomputing, 2016, 174: 651-660", " Math-data integrated prediction model for ship maneuvering motion. Ocean Engineering, 2023, 285: 115255".

I have some thoughts that I consider crucial and on which I want to see the opinion / comments of the authors on the final manuscript.

4.      The representation of the additional mass coefficients in Eq. (1) is questionable. Linear and angular accelerations represent the fluid inertial forces and moments generated due to the ship's deviation from its navigational state. These forces and moments always try to impede the non-inertial motion of the ship in the relevant direction. Therefore, ${x_{\dot u}} < 0$ and ${m_x} = - {x_{\dot u}}$. The remaining terms are similar.

5.      In this paper, Eqs. (2)-(4) are used as a revised version of the Inoue model. From which reference? Or why is it written that way?

6.      How is the regularization factor chosen C when using truncated LS-SVM parameter identification?

7.      What are the inputs and outputs of the truncated LS-SVM for parameter identification of ships?

8.      It is known that the larger the number of data in the training set, and the results will be more accurate. What is the purpose of discussing the size of the data in this paper? Or does it have any particular impact on ship motion prediction? Is it possible to summarize the most appropriate data size criteria for the truncated LS-SVM?

Author Response

We are truly grateful for the comments and the work on our manuscript from the editor and reviewers. The comments helped us improve the quality of the manuscript. Based on these comments and suggestions, we have made modifications to the original manuscript. The point to point response to the editor and reviewers’ comments is shown below.

In this paper, the main research is divided into the following aspects: verifying the accuracy and robustness of the truncated LS-SVM using test data of different sizes; demonstrating the stability of the parameters and the decrease in the confidence interval range due to the increase in the number of data; quantifying the performance of the parameters by three metrics and verifying the correctness of the parameters by using the surge, sway force, and yaw moments in the PMM test.

1.In Section 5, Fig 4, all vertical coordinates are replaced with nondimensional parameters.

[RE]: Thanks for the remind, the new Fig 4, Fig5 and Fig 6 are replotted with the nondimensional variables as suggested, Please check the new version.

2.It would be highly more convenient for the reader if they add a table with all abbreviations and their meaning in the first before the introduction section. It was very difficult to keep track as it is now.

[RE]: The table with all abbreviations and their meaning is added in the new version

3.Further enrichment of the article, please refer to the following papers: " Large tanker motion model identification using generalized ellipsoidal basis function-based fuzzy neural networks. IEEE Transactions on Cybernetics, 2015, 45(12): 2732-2743", " Hybrid recursive least squares algorithm for online sequential identification using data chunks. Neurocomputing, 2016, 174: 651-660", " Math-data integrated prediction model for ship maneuvering motion. Ocean Engineering, 2023, 285: 115255".

I have some thoughts that I consider crucial and on which I want to see the opinion / comments of the authors on the final manuscript.

[RE]: Thanks for the recommendation, The above references are very interesting and close to the topic in this paper. They are cited in the paper.

4.The representation of the additional mass coefficients in Eq. (1) is questionable. Linear and angular accelerations represent the fluid inertial forces and moments generated due to the ship's deviation from its navigational state. These forces and moments always try to impede the non-inertial motion of the ship in the relevant direction. Therefore, ${x_{\dot u}} < 0$ and ${m_x} = - {x_{\dot u}}$. The remaining terms are similar.

[RE]: In the current version, the manoveruing model is given as

5.In this paper, Eqs. (2)-(4) are used as a revised version of the Inoue model. From which reference? Or why is it written that way?

[RE]: It is the revised version of the Inoue model, indeed, it was simplified and firstly given during the manovuering course in our center and I think some information can also be found in Sutulo, S.; Guedes Soares, C. Development of a Core Mathematical Model for Arbitrary Manoeuvres of a Shuttle Tanker. Appl. Ocean Res. 2015, 51, 293–308. https://doi.org/10.1016/j.apor.2015.01.008.

6.How is the regularization factor chosen C when using truncated LS-SVM parameter identification?

[RE]: It was kept constant and we choose it as 10^2, and indeed, this factor also play an important roles, in each identification, it should be optimised. In this paper, in order to compare the effect of training data size, we choos it as a constant value.

7.What are the inputs and outputs of the truncated LS-SVM for parameter identification of ships?

[RE]: The input is the surge speed, sway speed, and yaw rate and it couped terms, as indicated in the manovuering model, the output is the surge forces, sway foces and yaw moments.

8.It is known that the larger the number of data in the training set, and the results will be more accurate. What is the purpose of discussing the size of the data in this paper? Or does it have any particular impact on ship motion prediction? Is it possible to summarize the most appropriate data size criteria for the truncated LS-SVM?

[RE]: we agree with this comment, It is common knowledge that the larger the number of data in the training set, and the results will be more accurate. But, for LS-SVM, it is not the case for the large data size. As suggested by Suykens et. al [54], the size of the training set for the least square support vector machine (LS-SVM) should be restricted below 2000 data points. In this paper, we proposed the truncated LS-SVM to solve this problem. From the results, we can find that the proposed algorithm can still provide a stable estimation even when the training set is 1000. It is a good idea to find the optimal data size critical for the truncated LS-SVM, we will consider this idea in the further work.

Suykens, J. A. K.; Van Gestel, T.; De Brabanter, J.; De Moor, B.; Vandewalle, J. Least Squares Support Vector Machines; World Scientific, 2002. https://doi.org/10.1142/9789812776655.

Reviewer 2 Report

Comments and Suggestions for Authors

1. This paper investigates the data-driven parameter estimation of a nonlinear ship 327 manoeuvring model, the basic idea is to use SVM methods. There exist a plenty of parameter estimation methods for linear systems and nonlinear systems in the area. Thus the authors should carefully analyze some relevant references.

2. The paper involves the SVM methods, thus the relevant SVM estimation methods should be cited in the paper such as Ma H. A novel multi-innovation gradient support vector machine regression method. ISA Trans. 2022; 130:343-359. Ma H. Multi-innovation Newton recursive methods for solving the support vector machine regression problems. Int J Robust Nonlinear Control. 2021; 31(15):7239-7260.

3,  The  paper contains some materials worthy of publication but the revision is necessary. 

4. The paper contains some grammatical errors. The authors should carefully check the whole paper and avoid them.

5. In fact, parameter estimation for linear systems and nonlinear systems have many identification methods. Thus some relevant methods should be surveyed and mentioned in the paper such as  the gradient-based methods, the least squares-based methods and Newton methods: Xu L. Parameter estimation for nonlinear functions related to system responses. Int J Control Autom Syst. 2023;21(6):1780-1792. Xu L.  Separable synchronous multi-innovation gradient-based iterative signal modeling from on-line measurements. IEEE Trans Instrum Meas. 2022; 71: 6501313. Xu L. Separable Newton recursive estimation method through system responses based on dynamically discrete measurements with increasing data length.  Int J Control Autom Syst. 2022; 20(2)432-443. Ding F. Least squares parameter estimation and multi-innovation least squares methods for linear fitting problems from noisy data. J Comput Appl Math. 2023; 426: 115107. 

4, The paper can be accepted after revision according to the above comments.

Comments on the Quality of English Language

3,  The  paper contains some materials worthy of publication but the revision is necessary. 

4. The paper contains some grammatical errors. The authors should carefully check the whole paper and avoid them.

Author Response

We are truly grateful for the comments and the work on our manuscript from the editor and reviewers. The comments helped us improve the quality of the manuscript. Based on these comments and suggestions, we have made modifications to the original manuscript. The point to point response to the editor and reviewers’ comments is shown below.

  1. This paper investigates the data-driven parameter estimation of a nonlinear ship 327 manoeuvring model, the basic idea is to use SVM methods. There exist a plenty of parameter estimation methods for linear systems and nonlinear systems in the area. Thus the authors should carefully analyze some relevant references.

[RE]: Thanks for the comments, the paper was revised carefully and more relevant references are reviewed. Please check it.

2 The paper involves the SVM methods, thus the relevant SVM estimation methods should be cited in the paper such as Ma H. A novel multi-innovation gradient support vector machine regression method. ISA Trans. 2022; 130:343-359. Ma H. Multi-innovation Newton recursive methods for solving the support vector machine regression problems. Int J Robust Nonlinear Control. 2021; 31(15):7239-7260.

[RE]: The recommented papers are interesting and cited in the new version.

3, The paper contains some materials worthy of publication but the revision is necessary.

[RE]: Thanks, please check the new version.

  1. The paper contains some grammatical errors. The authors should carefully check the whole paper and avoid them.

[RE]: We thoughtly check the whole paper and correct the grammatical errors.

  1. In fact, parameter estimation for linear systems and nonlinear systems have many identification methods. Thus some relevant methods should be surveyed and mentioned in the paper such as  the gradient-based methods, the least squares-based methods and Newton methods: Xu L. Parameter estimation for nonlinear functions related to system responses. Int J Control Autom Syst. 2023;21(6):1780-1792. Xu L.  Separable synchronous multi-innovation gradient-based iterative signal modeling from on-line measurements. IEEE Trans Instrum Meas. 2022; 71: 6501313. Xu L. Separable Newton recursive estimation method through system responses based on dynamically discrete measurements with increasing data length.  Int J Control Autom Syst. 2022; 20(2)432-443. Ding F. Least squares parameter estimation and multi-innovation least squares methods for linear fitting problems from noisy data. J Comput Appl Math. 2023; 426: 115107. 

[RE]: The recomented papers are interesting and cited in the current version.

4, The paper can be accepted after revision according to the above comments.

[RE]: Thanks for your comments, please check the new version.

Reviewer 3 Report

Comments and Suggestions for Authors

The presented paper addresses an important aspect of hydrodynamic coefficient estimation using a data-driven approach, specifically the truncated LS-SVM method. The utilization of experimental data from a shallow water towing tank adds practical relevance to the study. The paper's focus on validating the proposed approach by varying the size of the training data is commendable, as it provides insights into its accuracy and robustness.The outcomes of this research, including the estimation of nondimensional hydrodynamic coefficients, assessment of parameter uncertainty through confidence intervals, and the validation against reference data using statistical measures, are all significant contributions. The discussion on the advantages of employing larger training sets for improved results is particularly insightful.

The paper effectively communicates the methodology, results, and implications of the study. However, further details on the experimental setup and data collection process would enhance the reader's understanding. Additionally, discussing potential limitations or challenges faced during the truncated LS-SVM implementation could provide a more comprehensive perspective.

Comments on the Quality of English Language

The quality of the English language still need polishing more as a scientific way. 

Author Response

We are truly grateful for the comments and the work on our manuscript from the editor and reviewers. The comments helped us improve the quality of the manuscript. Based on these comments and suggestions, we have made modifications to the original manuscript. The point to point response to the editor and reviewers’ comments is shown below.

The presented paper addresses an important aspect of hydrodynamic coefficient estimation using a data-driven approach, specifically the truncated LS-SVM method. The utilization of experimental data from a shallow water towing tank adds practical relevance to the study. The paper's focus on validating the proposed approach by varying the size of the training data is commendable, as it provides insights into its accuracy and robustness.The outcomes of this research, including the estimation of nondimensional hydrodynamic coefficients, assessment of parameter uncertainty through confidence intervals, and the validation against reference data using statistical measures, are all significant contributions. The discussion on the advantages of employing larger training sets for improved results is particularly insightful.

The paper effectively communicates the methodology, results, and implications of the study. However, further details on the experimental setup and data collection process would enhance the reader's understanding. Additionally, discussing potential limitations or challenges faced during the truncated LS-SVM implementation could provide a more comprehensive perspective.

[RE]: We greatly appreciate your time and effort on this draft, meanwhile, the positive comment on this draft. The details on the experimental setup and data collection process were included in this version. and the potential limitations or challenges for using truncated LS-SVM are also discussed in the conclusion. Please check the new version.
